# Role of Myeloid Cell-Specific TLR9 in Mitochondrial DNA-Induced Lung Inflammation in Mice

**DOI:** 10.3390/ijms24020939

**Published:** 2023-01-04

**Authors:** Kris Genelyn Dimasuay, Bruce Berg, Niccolette Schaunaman, Hong Wei Chu

**Affiliations:** Department of Medicine, National Jewish Health, 1400 Jackson Street, Room A639, Denver, CO 80206, USA

**Keywords:** mitochondrial DNA, lung, macrophages, inflammation, palmitic acid

## Abstract

Mitochondrial dysfunction is common in various pathological conditions including obesity. Release of mitochondrial DNA (mtDNA) during mitochondrial dysfunction has been shown to play a role in driving the pro-inflammatory response in leukocytes including macrophages. However, the mechanisms by which mtDNA induces leukocyte inflammatory responses in vivo are still unclear. Moreover, how mtDNA is released in an obese setting has not been well understood. By using a mouse model of TLR9 deficiency in myeloid cells (e.g., macrophages), we found that TLR9 signaling in myeloid cells was critical to mtDNA-mediated pro-inflammatory responses such as neutrophil influx and chemokine production. mtDNA release by lung macrophages was enhanced by exposure to palmitic acid (PA), a major saturated fatty acid related to obesity. Moreover, TLR9 contributed to PA-mediated mtDNA release and inflammatory responses. Pathway analysis of RNA-sequencing data in TLR9-sufficient lung macrophages revealed the up-regulation of axon guidance molecule genes and down-regulation of metabolic pathway genes by PA. However, in TLR9-deficient lung macrophages, PA down-regulated axon guidance molecule genes, but up-regulated metabolic pathway genes. Our results suggest that mtDNA utilizes TLR9 signaling in leukocytes to promote lung inflammatory responses in hosts with increased PA. Moreover, TLR9 signaling is involved in the regulation of axon guidance and metabolic pathways in lung macrophages exposed to PA.

## 1. Introduction

Mitochondrial dysfunction, characterized by decreased ATP production, membrane potential and expression of mitochondrial complexes I, III, IV, but increased reactive oxygen species (ROS) [1], has been observed in inflamed lungs of various diseases such as asthma [2,3,4], chronic pulmonary obstructive disease (COPD), pulmonary fibrosis [5] and acute lung injury [6]. One of the biomarkers for mitochondrial dysfunction during the inflammatory process is increased release of mitochondrial DNA (mtDNA). For example, extracellular mtDNA in acute lung injury has been demonstrated to promote lung inflammation [7,8]. The mechanisms by which mtDNA is released remain unclear, but multiple pathological factors including infection, injury and inflammation are associated with mtDNA release.

Recently, obesity, one of the major health risk factors, has been linked to mitochondrial dysfunction [9,10]. The role of mtDNA in obese asthma has not been well studied. Cell-free mtDNA increased in the plasma of patients with type 2 diabetes, which was associated with oxidative stress and insulin resistance in skeletal muscle tissue [11]. How an obese condition may induce mitochondrial dysfunction such as increased mtDNA content or release remains poorly understood. Metabolic abnormality such as excessive saturated fatty acids is a major feature of obesity. Palmitic acid (PA), the most abundant fatty acid in the human body, has been shown to be increased in obese subjects [12]. However, whether PA may contribute to mtDNA release has not been reported. One of the major goals of our current study was to investigate if PA induces the release of mtDNA from lung macrophages, a key cell type involved in the innate or inflammatory response of the lung. 

In our previous study, we demonstrated that administration of mtDNA amplified airway inflammation in allergic mice [13]. mtDNA may induce the inflammatory response using several signaling mechanisms including Toll-like receptor 9 (TLR9). TLR9 is localized to endosomes and lysosomes, and contributes to the host defense response by recognizing CpG motifs in DNA from viruses and bacteria [14,15] as well as host mtDNA [16]. The mtDNA/TLR9 signaling axis has been implicated in diseases such as rheumatoid arthritis [17], atherosclerosis [18], nonalcoholic fatty liver disease [19] and idiopathic pulmonary fibrosis [20]. Murakami et al. [21] demonstrated that TLR9 exacerbates airway inflammation in an allergic asthma model using whole body TLR9-deficient mice. However, whether TLR9 deficiency in myeloid cells including macrophages contributes to mtDNA-mediated lung inflammation has not been explored. Here, we used our recently generated myeloid-specific TLR9-deficient mice to determine the in vivo role of TLR9 signaling in myeloid cells during mtDNA-induced lung inflammation. Furthermore, we examined the role of PA in mtDNA release and the inflammatory response in TLR9-sufficient and -deficient lung macrophages to determine the potential mechanisms underlying the pro-inflammatory effect of PA.

## 2. Results

### 2.1. Lung Inflammation Induced by mtDNA Was Inhibited in Myeloid Cell-Specific TLR9 Knockout Mice

To determine the role of TLR9 in myeloid lineages such as monocytes and macrophages in mtDNA-induced lung inflammation, we used tamoxifen-inducible TLR9 CKO mice that express Cre recombinase under the control of the lysozyme M (Lyz2) promoter. The Lyz2 is used for gene deletion of myeloid lineages such as monocytes and macrophages [22]. First, under TE buffer treatment, we confirmed about a 65% decrease in TLR9 mRNA in isolated lung macrophages from the CKO mice (TLR9 mRNA relative level, 0.94 ± 0.07) compared to TLR9 sufficient (wild-type, WT) mice (1.55 ± 0.16) (Figure 1A). mtDNA significantly upregulated TLR9 mRNA in lung macrophages isolated from WT mice but not from CKO mice. mtDNA increased the number of neutrophils (Figure 1B) and macrophages (Figure 1C) in BALF from WT mice. Notably, TLR9 deficiency in myeloid cells significantly reduced the recruitment of neutrophils and macrophages after mtDNA treatment (Figure 1B,C). There was a decrease in the levels of neutrophilic chemoattractant LIX in the BALF of TLR9 CKO mice compared to WT mice (Figure 1D). Importantly, production of the chemokine KC (CXCL1) was inhibited in cultured macrophages from the TLR9 CKO vs. WT mice that were treated in vivo with mtDNA (Figure 1E). This suggests that TLR9 in myeloid cells such as macrophages is critical to mtDNA-induced lung inflammation. 

### 2.2. PA Treatment In Vivo Increased Lung Inflammation

Our data in Figure 1 clearly demonstrates that mtDNA was able to induce lung inflammation in mice. PA is known to induce the pro-inflammatory response in macrophages [23], but whether this is associated with the in vivo lung release of mtDNA and inflammation has not been investigated. Using a wild-type mouse model of PA treatment via oropharyngeal administration, we observed an increased number of neutrophils (Figure 2A) and LIX (Figure 2B) compared to mice treated with BSA. Similarly, PA also increased macrophages in BALF of wild-type mice (Figure 2C). Unexpectedly, mtDNA in BALF was decreased in PA-treated mice (Figure 2D), which was accompanied by increased DNase activity (Figure 2E).

### 2.3. Macrophages Stimulated with PA Increased the Release of mtDNA and Pro-Inflammatory Cytokine Production

The reduced mtDNA levels in the BALF of mice treated with PA in Figure 2 may be due to DNA degradation by increased DNase activity under an inflammatory milieu. However, it is not clear whether macrophages stimulated a PA decrease or increased mtDNA release. Therefore, we isolated macrophages from the lungs of WT mice and stimulated them with PA. We found that PA increased the release of mtDNA (Figure 3A), as well as pro-inflammatory cytokine KC (Figure 3B). To confirm that PA was processed by macrophages, an Oil Red O staining was performed to detect lipid droplets. PA-stimulated macrophages had stronger red staining inside the cells than the control cells, indicating PA was processed into lipid droplets (Figure 3C). 

### 2.4. TLR9 Deficiency in Macrophages Attenuated In Vivo mtDNA Release and Inflammatory Response following PA Treatment

Having shown that PA increased mtDNA release from lung macrophages and mtDNA treatment in vivo induced TLR9 expression in lung macrophages, we sought to determine if TLR9 in macrophages may in turn regulate or amplify PA-mediated mtDNA release. Lung macrophages isolated from TLR9 CKO or TLR9-sufficient mice with or without in vivo PA treatment were cultured for 24 h (no further stimulation in vitro). TLR9 sufficient macrophages from mice treated with PA (vs. BSA) increased mtDNA release (Figure 4A). Importantly, macrophages with TLR9 deficiency did not increase mtDNA release after PA treatment, which was consistent with the KC release in the supernatant (Figure 4B). 

### 2.5. Pathway Analysis of Bulk RNA-Seq Data in TLR9-Sufficient and -Deficient Mouse Lung Macrophages

To reveal the potential mechanisms by which TLR9 signaling in macrophages modulated the inflammatory response to PA, we performed bulk RNA-sequencing in lung macrophages isolated from TLR9 CKO and TLR9-sufficient mice treated with or without PA, and analyzed the transcriptomic changes of genes and associated pathways. TLR9 expression levels in macrophages from TLR9 CKO mice treated with BSA were reduced by about 4-fold as compared to the cells from TLR9-sufficient mice treated with BSA (Figure 5), which was consistent with the TLR9 qPCR data shown in Figure 1A. We focused on the genes changed by PA in TLR9-sufficient macrophages, and then examined the effect of TLR9 deficiency on PA-mediated transcriptomic changes in lung macrophages. In TLR9-sufficient macrophages (*n* = 8), PA treatment as compared to the control (BSA treatment, *n* = 4) significantly upregulated 1,161 genes, while it downregulated 512 genes (Figure 6A). Table 1 illustrates the gene pathways of those differentially expressed genes. As expected, multiple cytokines were upregulated by PA in TLR9-sufficient macrophages (Table 2). Notably, axon guidance, a pathway initially described for neuron communication [24], was significantly upregulated by PA. Of note, axon guidance has been recently shown to participate in inflammation in metabolic disorders and other pathological conditions such as lipopolysaccharides (LPS) treatment in the lung [25,26]. Meanwhile, the genes downregulated by PA in TLR9-sufficient lung macrophages were those in the metabolic pathways. Interestingly, PA-stimulated lung macrophages from TLR9-deficient (*n* = 8) vs. -sufficient (*n* = 8) mice increased the expression of 382 genes with those associated with metabolic pathways being the most significant (Figure 6B, Table 3 and Table 4). There were 370 genes downregulated in PA-stimulated TLR9-deficient vs. -sufficient macrophages. Pathway analysis revealed only two gene pathways with *p* < 0.05 (Table 3 and Table 4) with the axon guidance as the major pathway downregulated in PA-treated TLR9-deficient lung macrophages. We used qPCR to validate the expression of Sema3b, one of the top hits, in isolated lung macrophages from which RNA samples were still available to us after the RNA-seq experiment. Lung macrophages from TLR9-deficient vs. TLR9-sufficient mice treated with PA demonstrated less Sema3b mRNA expression (Sema3b mRNA median levels, 0.5 vs.13.4, *p* = 0.05), supporting our RNA-seq data shown in Table 4.

## 3. Discussion

Recent research in investigating how obesity is linked to inflammation has suggested a role of mitochondrial dysfunction. However, the interactions of risk factors associated with obesity, mitochondrial dysfunction and inflammation remain largely unclear. By using our newly generated conditional myeloid cell TLR9 knockout mouse model, we demonstrated the pivotal role of lung macrophage TLR9 signaling in driving inflammation in lungs exposed to PA and mtDNA.

PA is a major saturated fatty acid in human body, which can be increased in an obese condition. Plasma saturated fatty acid levels are positively correlated with airway neutrophilic inflammation in obese asthma patients [27]. However, a previous mouse model study did not demonstrate the ability of PA to induce airway neutrophilic inflammation when it was delivered intraperitoneally [28]. Here, we were able to show that airway (oropharyngeal route) delivery of PA in mice significantly increased the recruitment of neutrophils into the lung. How PA in the lung increases the inflammatory response has not been well understood. In the current study, we found that PA increased the release of mtDNA directly from macrophages, which may in part be utilized by TLR9 signaling to induce the pro-inflammatory response. It has been well established that PA induces the inflammatory response in macrophages [29,30], but few studies have used a lung macrophage culture model to study how PA regulates lung inflammation. We found that lung macrophages primed in vivo with PA or stimulated in vitro with PA increased the release of mtDNA and neutrophil chemokine KC. mtDNA has been suggested to mediate the inflammatory response through several signaling pathways including TLR9, cGAS/cGAMP/STING and the inflammasome pathways [31,32]. As mtDNA contains unmethylated CpG motifs, we investigated TLR9 signaling. Our study represents the first study to examine the in vivo role of TLR9 from myeloid cells in mtDNA-mediated lung inflammation. Our data showed that lung neutrophil inflammation with TLR9 deficiency in myeloid cells was about 3-fold lower than that in TLR9 sufficient mice (Figure 1B), suggesting the major contribution of TLR9 signaling in myeloid cells to mtDNA-mediated lung inflammation. 

Although the mechanisms by which PA induces inflammation has been relatively well studied, there are few studies using the RNA-seq approach to reveal potential new mechanisms. Our RNA-seq data in lung macrophages suggested several interesting pathways involved in the PA-mediated inflammatory responses. Notably, axon guidance was shown to be the top pathway that was significantly up-regulated by PA in TLR9-sufficient lung macrophages. Axon guidance signaling was initially studied in neurons as a critical pathway to regulate neural development such as axon growth and guidance [24]. Recently studies have suggested the role of axon guidance such as semaphorins (sema) and their receptors (e.g., plexins) in regulating innate immunity and inflammation [26]. Sema/plexin axes have been shown to regulate macrophage and neutrophil functions and contribute to several lung diseases including acute lung injury, asthma, COPD and pulmonary fibrosis [33]. Although semaphorins have been linked to metabolic disorders [34], whether saturated fatty acids such as PA regulate the expression of semaphorins/plexins remains unknown. In this study, we found that PA up-regulated multiple semaphorins/plexins in macrophages from the wild-type mice. Importantly, TLR9-deficient versus sufficient lung macrophages decreased the expression of semaphorins/plexins along with chemokine production following PA treatment. Our data suggest that induction of axon guidance signaling may be a novel mechanism whereby PA enhances the pro-inflammatory response associated with TLR9 signaling. 

There are several additional interesting findings in this study. First, genes in the metabolic pathways were down-regulated by PA in lung macrophages from the wild-type mice, but were up-regulated by TLR9 deficiency in PA-stimulated cells. Whether down-regulated genes in metabolic pathways contribute to the pro-inflammatory effect of PA in our current study remains unclear. Metabolic reprogramming in macrophages has been proposed to play a significant role in various lung diseases such as asthma [35]. In our study, several mitochondria-related genes such as Mt-ATP6, Mt-Co2 and Mt-Co3 involved in maintaining mitochondrial functions [36], and Arg2 (arginase 2) were reduced by PA, suggesting a damaging effect of PA on mitochondrial function. Arginase 2 has been shown to promote the resolution of inflammation in lung macrophages [37]. We speculate that modulation of metabolic pathways by TLR9 signaling may also contribute to the pro-inflammatory function of PA. An intriguing finding is about the fact that PA increased mtDNA release from cultured macrophages, but decreased the levels of mtDNA found in BAL fluid of PA-treated wild-type mice. The exact mechanism for this observation is not clear, but increased levels of DNase we observed in mice with lung neutrophilic inflammation may contribute to this finding. 

There are several limitations to the current study. First, we used PA as an experimental agent to study the acute impact of metabolites in an obese condition on lung inflammation. To better mimic the obese condition, a chronic model of mice fed with high-fat diet will be considered in our future experiments. Second, some of the novel findings from our bulk RNA-seq experiments warrants validation by functional assays or experiments. For example, the role of semaphorins/plexins in TLR9-mediated mtDNA signaling could be further explored. Lastly, due to the limited number of macrophages from an individual mouse lung, we were not able to perform Western blot, after the cells were used for RNA extraction, to confirm TLR9 protein reduction in macrophages from myeloid-specific TLR9 conditional knockout mice. 

In summary, mitochondrial dysfunction is a common feature in multiple lung pathological conditions including obesity. Understanding the role of the mtDNA/TLR9 axis in lung inflammation induced by saturated fatty acids may provide insights into new mechanisms of various human diseases associated with obesity, which may eventually provide novel therapeutic targets.

## 4. Materials and Methods

### 4.1. Preparation and Quantification of mtDNA

mtDNA was prepared and quantified according to our previous publication [13]. Briefly, mtDNA was isolated from the lungs of wild-type (WT) mice on a C57BL/6 background using a mtDNA Isolation Kit (Biovision, Waltham, MA, USA), and sonicated using a Vibracell ultrasonic processor (Sonics and Materials, Inc., Newtown, CT, USA). For quantification of mtDNA, mtDNA was extracted from cell-free bronchoalveolar lavage fluid (BALF) using a DNA mini kit (Qiagen) according to the manufacturer’s protocol. Mouse mitochondrial specific 16S rRNA gene was used for quantification of mtDNA by real-time PCR with a purified mtDNA standard curve. 

### 4.2. Generation of Inducible Myeloid-Specific TLR9 Conditional Knockout (CKO) Mice

All animal studies were approved by the Institutional Animal Care and Use Committee (IACUC) at National Jewish Health (protocol# AS2972-03-23). LysM-Cre ERT2 (tamoxifen-inducible Cre expression) mice were purchased from the Jackson Laboratory. TLR9 floxed (TLR9^fl/fl^) mice, a kind gift from Dr. Shlomchik at the University of Pittsburgh, were generated as described previously [38]. Both strains of mice were crossed and inbred for at least six generations to produce LysM-Cre+ (CKO) or LysM Cre- (wild-type, WT) TLR9^fl/fl^ mice. Mice were fed with tamoxifen chow for 7 d to induce the expression of Cre recombinase for the specific deletion of TLR9 in myeloid cells. Then, the mice were fed with regular chow for another 7 d before treatment with mtDNA or palmitic acid. 

### 4.3. Mouse Models of mtDNA and Palmitic Acid Treatments

To determine if TLR9 in myeloid cells contributes to mtDNA-induced inflammation, TLR9 CKO and WT mice were administered with sonicated mtDNA in Tris-EDTA (TE buffer) at 5 µg/mouse or 50 µL TE buffer through oropharyngeal administration. 

To test if PA induces mtDNA release, WT BALB/c mice or TLR9 CKO and WT mice on C57BL/6 background were challenged via oropharyngeal inoculation with PA at 100 µM/mouse conjugated with 0.1% fatty acid-free bovine serum albumin (BSA) or 0.1% BSA (control) once daily for three consecutive days. Our mouse model study using PA at 100 µM/mouse was based on previous publications. In a study published in Nat Commun [39], mice received PA injection at 5 mM in 500 µL solution via the tail vein twice a day for 7 days to induce myocardial inflammation. A previous study [40] tested the lung distribution of tail vein injected radio-labelled nanoparticles, and found that 2 to 5% of the injected dose of nanoparticles was distributed to the lung 2 to 5 days after the injection. Given the results from these two publications, we decided to deliver PA to the mouse lung at 100 µM (2% (distribution to the lung) × 5 mM PA (if injected via a tail vein) = 100 µM) for our mouse experiment.

Mice were sacrificed after 24 h of mtDNA treatment or the last PA challenge. BALF was collected for cell differential counts while cell-free BALF was used for measuring cytokines and mtDNA using ELISA and qPCR, respectively. Lungs were harvested for macrophage isolation and culture. 

### 4.4. Isolation and Culture of Macrophages from Mouse Lung Tissue

Excised mouse lung tissues from different treatment groups were digested using DMEM (Dulbecco’s modified Eagle’s Medium) with 200 µg/mL DNase1, 2 mg/mL collagenase and penicillin/streptomycin and incubated at 37 °C for 30 min. The tissues were passed through a 200 µM strainer followed by a 40 µM cell strainer. The collected suspension was lysed for red blood cells before they were resuspended in DMEM with 10% fetal bovine serum (FBS) and penicillin/streptomycin and plated in 48-well plates at 4 × 10^4^ cells/well. The medium was replaced after 2 h with fresh medium to isolate adhered macrophages. Cells were cultured for an additional 24 h before harvest. Cell culture supernatants were collected for ELISA and cells were saved in RLT lysis buffer (Catalog number: 79216, QIAGEN, Inc., Germantown, MD, USA) for RNA extraction and reverse transcription-polymerase chain reaction (RT-PCR). 

### 4.5. ELISA

Mouse LIX/CXCL5 and KC/CXCL1 were measured using a Duoset ELISA kit (R&D Systems, Minneapolis, MN, USA) according to manufacturer’s instructions. 

### 4.6. Quantitative Real-Time RT-PCR

RNA was extracted using the GenCatch Total RNA Extraction System (Epoch Life Sciences, Inc., Missouri City, TX, USA) according to manufacturer’s instruction, and was reverse transcribed to cDNA. The Taqman TLR9 qPCR assay was obtained from Applied Biosystems (Foster City, CA, USA). To calculate mRNA relative expression levels, the comparative cycle of threshold (ΔΔCT) method was used with the housekeeping gene 18S rRNA as an internal control.

### 4.7. Bulk RNA Sequencing and Analysis

RNA extracted from isolated lung macrophages was used for bulk RNA sequencing. RNA samples with good quantity and quality for RNA-seq were from TLR9 CKO mice treated with BSA (*n* = 7) or PA (*n* = 8), and TLR9-sufficient mice treated with BSA (*n* = 4) or PA (*n* = 8). Preparation of the RNA library, transcriptome sequencing and downstream analyses were conducted by Novogene Co., Ltd. (Beijing, China). The RNA library was generated by using the NEBNext Ultra II RNA library prep kit (New England BioLabs Inc., Ipswich, MA, USA) with a poly-A mRNA selective workflow (non-directional, no rRNA depletion step). Each RNA sample was then sequenced using Illumina PE150 technology with a target output of 6Gb PE150 data (20M PE150 read-pairs/40M individual 150bp reads). Genes with *p* value < 0.05 and log2-fold change >0 between the two groups were assigned as differentially expressed, and were included for pathway analysis using the functional enrichment analysis of the Database for Annotation, Visualization and Integrated Discovery (DAVID) Bioinformatics Resources, which was developed by the Laboratory of Human Retrovirology and Immunoinformatics (LHRI) in collaboration with the National Institute of Allergy and Infectious Diseases (NIAID).

### 4.8. Statistical Analyses

Data were presented as the mean and the statistical significance was determined using one-way ANOVA analysis with Holm–Sidak’s post hoc test for multiple comparisons. For two-group comparisons, a Student’s *t*-test was performed. A *p* < 0.05 was considered statistically significant.

## Figures and Tables

**Figure 1 ijms-24-00939-f001:**
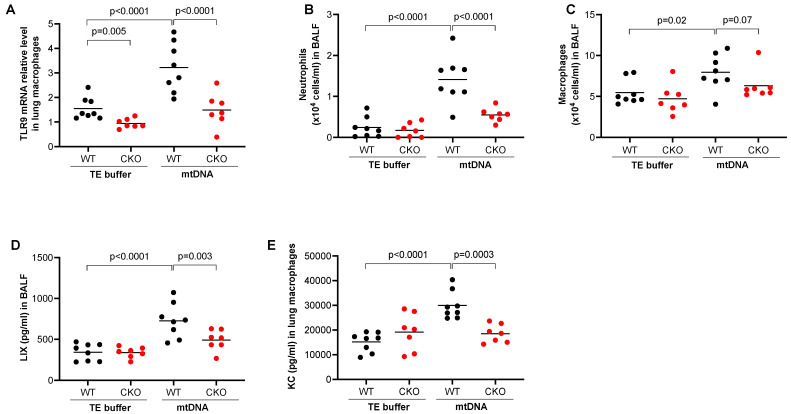
Deficiency of TLR9 in myeloid cells reduced lung inflammatory responses to mtDNA. After receiving the tamoxifen food, Cre^–^ (wild-type, WT) and Cre^+^ (conditional knockout, CKO) TLR9 floxed mice were treated oropharyngeally with mouse lung tissue-derived mtDNA prepared in Tris-EDTA (TE) buffer or TE buffer alone (control) for 24 h to obtain bronchoalveolar lavage fluid (BALF) and lung tissue for isolation of lung macrophages. (**A**) TLR9 mRNA levels as analyzed by qRT-PCR in isolated lung macrophages from WT and TLR9 CKO mice; (**B**) Neutrophil count in BALF; (**C**) Neutrophil chemoattractant LIX levels in BALF; (**D**) Macrophage count in BALF; (**E**) KC measured in the supernatants of macrophages isolated from mice treated with TE or mtDNA. Macrophages were cultured without further stimulation in vitro for 24 h before supernatants were collected for KC measurement. *n* = 7 to 8 mice per group.

**Figure 2 ijms-24-00939-f002:**
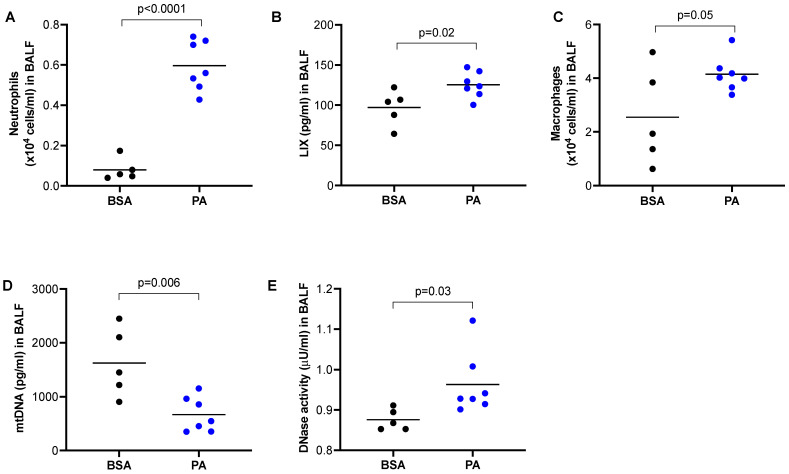
PA promoted lung inflammatory responses in mice. Wild-type BALB/c mice were treated oropharyngeally with PA prepared in BSA or BSA (control) for 24 h to obtain BALF. (**A**) Neutrophil count in BALF; (**B**) Neutrophil chemoattractant LIX levels in BALF; (**C**) Macrophage count in BALF; (**D**) mtDNA levels in BALF; (**E**) DNase activity levels in BALF. *n* = 5 to 7 mice per group.

**Figure 3 ijms-24-00939-f003:**
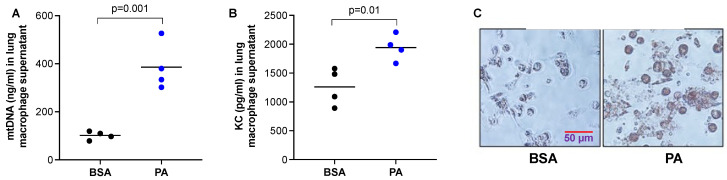
PA enhanced mtDNA and pro-inflammatory cytokine release from lung macrophages. Lung macrophages isolated from wild-type C57/BL6 mice were cultured and treated with PA (1 mM) and prepared in 0.1% BSA or 0.1% BSA (control) for 24 h to obtain the supernatants for measuring mtDNA (**A**) and neutrophil chemoattractant KC (**B**). (**C**) Oil red O staining in macrophages plated on a 24-well plate that were treated with PA prepared in BSA or BSA (control) for 24 h. *n* = 4 replicates per group.

**Figure 4 ijms-24-00939-f004:**
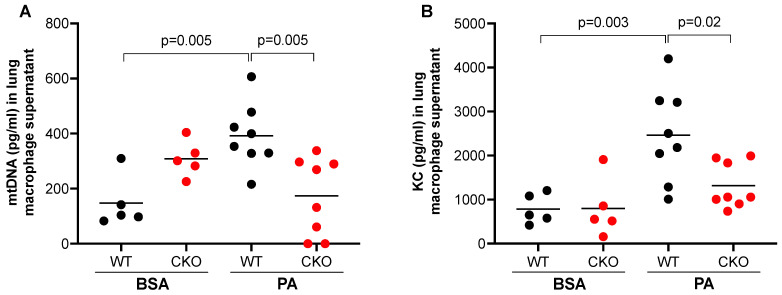
TLR9 deficiency reduced mtDNA and pro-inflammatory cytokine release from lung macrophages. PA enhanced the release of mtDNA (**A**) and neutrophil chemoattractant KC (**B**) from lung macrophages isolated from Cre^–^ (wild-type, WT), but not from Cre^+^ (conditional knockout, CKO) TLR9 floxed mice that were treated oropharyngeally with PA prepared in BSA or BSA (control) for 24 h. Macrophages were cultured without further stimulation in vitro for 24 h before supernatants were collected for mtDNA and KC measurement. *n* = 5 to 8 mice per group.

**Figure 5 ijms-24-00939-f005:**
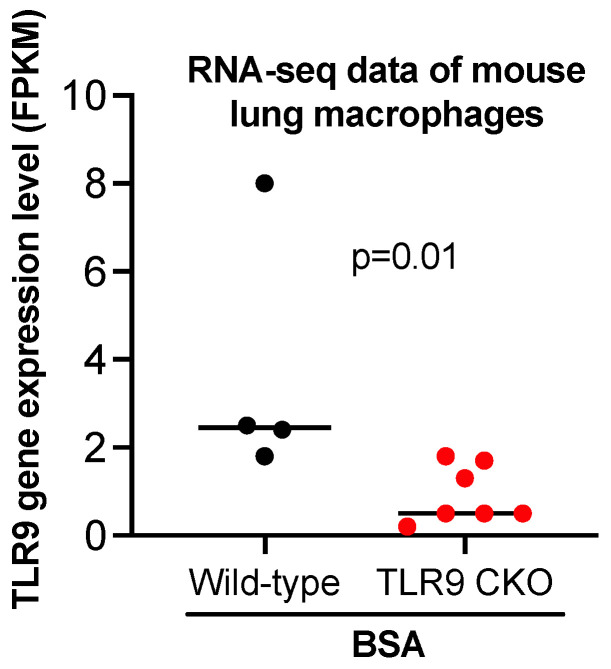
Bulk RNA-sequencing data showing reduced TLR9 gene expression in lung macrophages isolated from TLR9 CKO mice treated with BSA as compared to the cells from TLR9-sufficient mice treated with BSA. FPKM stands for fragments per kilobase of transcript per million mapped reads, indicating a relative expression of a gene proportional to the number of cDNA fragments of origin. *n* = 4 to 7 mice per group.

**Figure 6 ijms-24-00939-f006:**
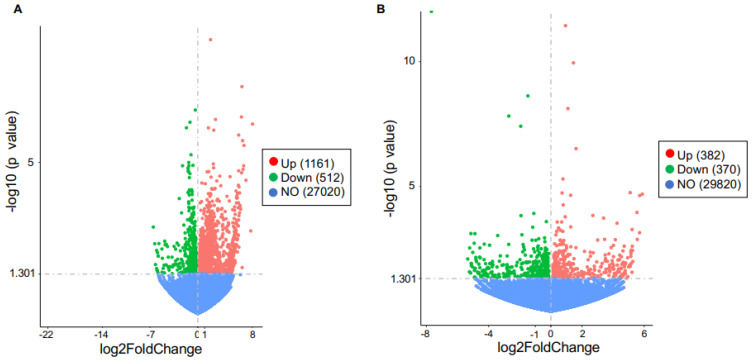
Volcano plot of differentially expressed genes regulated by in vivo PA treatment in TLR9-sufficient and -deficient mouse lung macrophages. (**A**) Genes changed (up- or down-regulated) or not changed (NO) in TLR9-sufficient macrophages treated with PA (*n* = 8) versus the BSA control (*n* = 4). (**B**) Genes changed (up- or down-regulated) or not changed (NO) in TLR9-deficient (*n* = 8) versus TLR9-sufficient (*n* = 8) macrophages treated with PA.

**Table 1 ijms-24-00939-t001:** Pathways of genes altered by palmitic acid (vs. BSA control) in wild-type mouse lung macrophages.

Signaling Pathways	Gene Count	%	Up- or Down-Regulated	*p* Value	False Discovery Rate (FDR)
Axon guidance	33	3.2	Up	4.8 × 10^−11^	1.3 × 10^−8^
Cell adhesion molecules	28	2.7	Up	8.0 × 10^−8^	1.1 × 10^−5^
PI3K-Akt signaling pathway	31	3.0	Up	1.4 × 10^−3^	7.7 × 10^−2^
Tight junction	18	1.7	Up	2.0 × 10^−3^	9.2 × 10^−2^
Leukocyte transendothelial migration	14	1.3	Up	3.3 × 10^−3^	1.3 × 10^−1^
Viral protein interaction with cytokine and cytokine receptor	12	1.1	Up	4.6 × 10^−3^	1.5 × 10^−1^
Cytokine–cytokine receptor interaction	25	2.4	Up	5.0 × 10^−3^	1.5 × 10^−1^
cGMP-PKG signaling pathway	17	1.6	Up	6.9 × 10^−3^	1.8 × 10^−1^
Vascular smooth contraction	15	1.4	Up	7.3 × 10^−3^	1.8 × 10^−1^
Focal adhesion	18	1.7	Up	1.3 × 10^−2^	2.4 × 10^−1^
Adherens junction	9	0.9	Up	1.7 × 10^−2^	2.4 × 10^−1^
Notch signaling pathway	8	0.8	Up	2.1 × 10^−2^	2.5 × 10^−1^
MAPK signaling pathway	18	4.1	Down	9.2 × 10^−5^	6.0 × 10^−3^
Toll-like receptor signaling pathway	10	2.3	Down	1.8 × 10^−4^	8.1 × 10^−3^
Metabolic pathways	53	12.2	Down	2.7 × 10^−4^	1.0 × 10^−2^
Chemokine signaling pathway	13	3.0	Down	5.1 × 10^−4^	1.7 × 10^−2^
TNF signaling pathway	8	1.8	Down	7.9 × 10^−3^	1.2 × 10^−1^
Ferroptosis	5	1.1	Down	8.4 × 10^−3^	1.2 × 10^−1^
Phagosome	10	2.3	Down	1.2 × 10^−2^	1.6 × 10^−1^
Mineral absorption	5	1.1	Down	2.3 × 10^−2^	2.3 × 10^−1^

**Table 2 ijms-24-00939-t002:** List of genes in selected pathways altered by palmitic acid in wild-type mouse lung macrophages.

Up-Regulated Genes by PA (*n* = 8) vs. BSA Control (*n* = 5)			
**Genes**	Log2 Fold Change	Genes	Log2 Fold Change
Axon Guidance		Cytokine–Cytokine Receptor Interaction	
Plxna2a2	0.5.50	IL-6	1.7
Plxna4a a4	0.9	CCL12	2.2
Plxnb1a lxnb1	1.0	CCR10	5.5
Sema3b	1.6	CXCL10	2.2
Sema3c	1.2	CXCL11	3
Sema3d	0.9	CXCL12	0.8
Sema3f	1.9	CXCL15	1.6
Sema3g	2.2	IL-15RA	1
Sema4c	1.5	IL5	3.7
Sema4f	2.1	CNTF	4.7
Sema6a	0.9	IL33	1.2
Sema6d	0.9	IL34	3
Sema7a	1.5	TNFSF10 (TRAIL)	1.2
Epha1	1.9	ISG20	1.8
Epha2	1.7	DR5 (TNFRSF10b)	0.6
Efna3	5.7	BMP3	2.5
Epha4	3.2	BMP2	1.5
Efna5	1.1	BMP5	3
Ephb1	5.1	BMPR1b	1.2
Enah	1.4	IL24	6
Nck2	0.9	LIFR	0.9
Nfatc4	1.1	ACKR3	1.2
Ptch1	0.5	BMPR1a	0.5
Fyn	0.3	Inhbb	2.5
Bmpr1b	1.2	IL17Rc	1.4
Cxcl12	0.8		
Shh	5.3	Leukocyte transendothelial migration	
Rgma	0.8	PECAM1	1.7
Robo2	1.5	Thy1	1.1
Robo3	5.5	VCAM1	1.2
Unc5b	1.3	Jam3	1.6
Unc5c	1.4	Esam	1.7
Wnt5a	2.6	Cldn5	2.3
		Cldn4	2.1
		Cldn3	2.2
		Cldn18	2.1
		Cxcl12	0.8
		Rapgef3	2.4
		Rapgef4	2.3
		Rapgef5	1.7
		Bcar1	1.5
**Down-regulated genes by PA (*n* = 8) vs. BSA control (*n* = 5)**			
Metabolic pathways			
Genes	Log2 fold change	Genes	Log2 fold change
Mt-ATP6	−1.3	C1galt1c1	−0.3
Large2	−1.8	Galc	−0.5
Ndufs3	−0.2	Aldh3a2	−0.2
Ndufc1	−0.5	Alas1	−0.3
Nme1	−0.3	Blvrb	−0.4
Setmar	−1	Bckdha	−0.6
Ugt1a7c	−0.8	Ckb	−0.7
B4galt6	−0.8	Mt-Co2	−1.4
Ugdh	−0.4	Mt-Co3	−1.2
Arg1	−1.2	Dgkz	−0.4
Ass1	−0.5	Dolk	−0.3
Gclm	−0.5	Galt	−0.5
Gstm1	−0.5	Gla	−0.3
Gstp2	−1.5	Ggcx	−0.4
Gsr	−0.3	Qrsl1	−0.5
Hmox1	−0.5	Mgst2	−0.6
Lipf	−0.7	Pnpla2	−0.3
Mgat4a	−0.4	Pigb	−0.5
Maoa	−0.5	Pik3cg	−0.4
Acyp1	−0.8	Pik3c2a	−0.4
Arg2	−1.3	Pip4k2a	−0.2
Peds1	−0.3	Pla2g4a	−0.2
Galnt6	−0.3	Pla2g7	−1.3
Pomk	−0.3	Sgms1	−0.5
Prune1	−0.3	Sgms2	−0.4
Synj2	−0.4	Tbxas1	−0.3
Xylt1	−0.8		

*p* values for comparisons of all the listed genes in this table are <0.05.

**Table 3 ijms-24-00939-t003:** Pathways of genes altered by TLR9 deficiency (vs. TLR9 sufficiency) in mouse lung macrophages treated with palmitic acid.

Signaling Pathways	Gene Count	%	Up- or Down-Regulated	*p* Value	False Discovery Rate (FDR)
Metabolic pathways	35	11.9	Up	1.5 × 10^−5^	2.5 × 10^−3^
Oxidative phosphorylation	8	2.7	Up	4.7 × 10^−4^	3.7 × 10^−2^
Diabetic cardiomyopathy	9	3.1	Up	1.5 × 10^−3^	7.8 × 10^−2^
Thermogenesis	9	3.1	Up	2.6 × 10^−3^	1.0 × 10^−1^
Prion disease	9	3.1	Up	6.4 × 10^−3^	2.0 × 10^−1^
Purine metabolism	6	2.0	Up	1.2 × 10^−2^	2.9 × 10^−1^
Axon guidance	11	3.7	Down	1.0 × 10^−4^	2.2 × 10^−2^
Ras signaling pathway	8	2.7	Down	3.1 × 10^−2^	1.0 × 10^0^

**Table 4 ijms-24-00939-t004:** List of genes in selected pathways altered by TLR9 deficiency (vs. TLR9 sufficiency) in mouse lung macrophages treated with palmitic acid.

Up-Regulated Genes by TLR9 Deficiency (*n* = 8) vs. TLR9 Sufficiency (*n* = 5)			
Genes	Log2 Fold Change	Genes	Log2 Fold Change
Metabolic pathways			
Mt-ATP6	1.2	Gmpr	0.3
Apip	0.4	Gucy2d	4.4
Atp5g3	0.2	Hexb	0.2
Ndst2	0.2	Lias	0.3
Ndufa12	0.7	Neu1	0.2
B3gnt6	5.2	Pisd	0.2
Acox2	2.0	Sdhb	0.1
Ampd2	0.2	Ptgs1	0.5
Aldh4a1	0.4	Ptges2	0.3
Alas2	2.0	Itpa	0.2
Alg3	0.2	Khk	0.3
Mt-Co3	0.9	Mgat4a	0.2
Cox11	0.6	Nudt5	0.2
Enpp3	0.8	Pld4	0.3
Gpt	0.7	Peds1	0.3
Qrsl1	0.4	Plod1	0.2
Gapdh	2.7	Uqcrfs1	0.2
Gamt	0.5		
**Down-regulated genes by TLR9 deficiency (*n* = 8) vs. TLR9 sufficiency (*n* = 5)**			
Axon guidance			
Genes	Log2 fold change	Genes	Log2 fold change
Plxna2a2	−0.750	Epha1	−1.3
Plxna4a a4	−0.6	Ephb1	−4.1
Plxnb2a lxnb1	−0.2	Robo3	−4.2
Sema3b	−1.1	Pik3r3	−0.8
Sema5a	−0.8	Arhgef12	−0.2
		Ptch1	−0.3

*p* values for comparisons of all the listed genes in this table are <0.05.

## Data Availability

The data presented in this study are available within the article. Raw RNA-sequencing data used for presenting data in this article are also provided in the Appendix A—Excel spreadsheets.

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
