# Peer review of "Role of Myeloid Cell-Specific TLR9 in Mitochondrial DNA-Induced Lung Inflammation in Mice"

_ijms, 2023, doi:10.3390/ijms24020939_

Round 1

Reviewer 1 Report

This is an interesting report linking TLR9 to mTDNA induced lung inflammation in mice. I have the following comments:

1. The link to obesity (treatment with 100µM PA) is not obvious. In obesity usually elevated levels of saturated lipids, but not free fatty acids at these high levels have been reported. 100µM PA is quite toxic to cells and also to mitochondria (induces uncoupling, PTP opening etc.). The authors need to better justify the used conditions and also mention potential side effects of this rather harsh treatment.

2. In the tables presenting the RNAseq data (Tab. 1-4) the FDR (false discovery rate) values should be provided.

3. The description of the RNAseq procedure is not sufficient.  How many samples were analysed? Which sequencing method was used (e.g. 3-prime sequencing or total sequencing after rRNA depletion)? Which coverages were achieved?

4. The top hits in the RNAseq should be verified by classical qPCR

Author Response

Please see the responses in the attached.pdf file.

Reviewer 2 Report

Major point:

Despite the authors have shown in Figure 1A that TLR9 mRNA level is reduced in lung macrophages isolated from TLR9 knockout vs. wild-type mice, this finding needs to be validated at the level of protein by Western blot.

Minor points:

1) Please provide all raw RNA-seq data as supplementary files.

2) Please change "TLR9 sufficient" to "TLR9-sufficient" (lines 17, 140, 213, 231, 233).

3) Please replace "genes, but" with "genes but" (line 19).

4) "Mitochondrial dysfunction, characterized by decreased ATP production, membrane potential and expression of mitochondrial complexes I, III, IV, but increased respiration and reactive oxygen species (ROS) [1], has been observed in inflamed lungs of various diseases such as asthma [2-4], COPD, pulmonary fibrosis [5] and acute lung injury [6]" (line 27) does not seem to be correct as human airway smooth muscle displayed reduced respiration in reference [1].

5) Please define the abbreviation for "COPD" (line 30), "LPS" (line 242).

6) Please replace "TLR9 deficient" with "TLR9-deficient" (lines 55, 247, 250, 312).

7) Please format "in vivo" using italics (line 58).

8) Please change "deficient" to "-deficient" (line 61).

9) Please replace "to" with "to the" (line 70).

10) Please change "Pittsburg" to "Pittsburgh" (line 77).

11) Please replace "days" with "d" (lines 80, 82, 91).

12) Please change palmitic acid (PA) with "PA" (line 88).

13) Please replace "fatty acid-free BSA" with "fatty acid-free bovine serum albumin" (line 90).

14) Please change "hours" to "hr" (lines 92, 103, 104, 159, 164, 183, 203, 205, 213, 224, 225, ).

15) Please replace "37ï‚°C" with "37 ï‚°C" (line 99).

16) Please change "minutes" to "min" (line 99).

17) Please replace "40μM" with "40 μM" (line 100).

18) Please provide either catalog number or composition of the "RLT Lysis buffer" mentioned in "Cell culture supernatants were collected for ELISA and cells were saved in RLT Lysis buffer for RNA extraction and RT-PCR" (line 104).

19) Please change "Lysis" to "lysis" (line 105).

20) Please replace "RT-PCR" with "real-time reverse transcription-polymerase chain reaction (RT-PCR)" (line 106).

21) Please provide city and state for the headquarters of "R&D Systems" mentioned in "Mouse LIX/CXCL5 and KC/CXCL1 were measured using a Duoset ELISA kit (R&D Systems) according to manufacturer’s instructions" (line 108).

22) Please change "reaction (RT-PCR)" to "reaction" (line 110).

23) Please provide city and state for the headquarters of "Epoch Life Sciences" mentioned in "RNA was extracted using the GenCatch Total RNA Extraction System (Epoch Life Sciences) according to manufacturer’s instruction, and was reversely transcribed to cDNA" (line 111).

24) Please provide city and state for the headquarters of "Applied Biosystems" mentioned in "The Taqman TLR9 qPCR assay was obtained from Applied Biosystems" (line 113) and in "List of genes in selected pathways altered by palmitic acid in wild-type (Cre [–]) mouse lung macrophages" (line 260?

25) Please replace "p<.05" with "P < 0.05" (line 129).

26) Please change "0.94±0.07" to "0.94 ± 0.07" (line 140).

27) Please replace "1.55±0.16" with "1.55 ± 0.16" (line 140).

28) Please change "mice, but" to "mice but" (line 142).

29) Please provide the figure reference for "Notably, TLR9 deficiency in myeloid cells significantly reduced the recruitment of neutrophils and macrophages after mtDNA treatment" (line 143).

30) Please change "of" to "of the" (line 147).

31) Please replace "(vs. WT)" with "vs. WT" (line 148).

32) Please replace "mitochondrial DNA (mtDNA)" with "mtDNA" (lines 156, 200, 221, 220).

33) Please change "bronchoalveolar lavage fluid (BALF)" to "BALF" (lines 159, 183).

34) Please replace "Neutrophil chemoattractant LIX levels in BALF; (D) Macrophage count in BALF" with "Macrophage count in BALF; (D) Neutrophil chemoattractant LIX levels in BALF" (line 162).

35) Please change "future" to "further" (lines 164, 225).

36) Please format "in vitro" using italics (lines 164, 225).

37) Please change "mice/group" to "mice per group" (line 165, 185, 226).

38) Please replace "Palmitic acid (PA)" with "Palmitic acid" (line 166).

39) "inflammation" could be changed to "inflammation in mice" (line 167).

40) Please change "activity" to "activity (Figure 2E)" (line 174).

41) Please change "Palmitic acid (PA)" to "Palmitic acid" (lines 181, 200, 221).

42) Please replace "bovine serum albumin (BSA)" with "BSA" (lines 182, 202, 205, 224, 255).

43) Please change "DNase" to "DNase activity" (line 189).

44) Please provide scale bar for at least one micrograph presented in Figure 3C.

45) From the legend of Figure 3C is not exactly clear what does "(-)" indicate?

46) Please replace "replicates/group" with "replicates per group" (line 205).

47) Please change "PA-mediated" to "palmitic acid-mediated" (line 207).

48) From "Having shown that PA increased mtDNA release from lung macrophages and mtDNA treatment in vivo induced TLR9 expression in macrophages, we sought to determine if TLR9 in macrophages may in turn regulate or amplify PA-mediated mtDNA release" (line 209) is not clear whether "mtDNA treatment in vivo induced TLR9 expression" also in lung macrophages?

49) Please replace "gene" with "genes" (line 233).

50) Please replace "n=8" with "N = 8" (lines 235, 244, 245, 254, 256 2x).

51) Please change "n=5" to "n = 5" (line 236, 255).

52) Please replace "while" with "while it" (line 236).

53) Please change "PA.Of" to "PA. Of" (line 240).

54) Please replace "as" with something like "as those stimulated by" (line 242).

55) Please replace "the" with something like "those of the" (line 243).

56) Although the authors claim that "Interestingly, PA-stimulated lung macrophages from TLR9-deficient (n=8, about 4-fold reduction of TLR9 expression) vs. sufficient (n=8) mice increased the expression of 382 genes with those associated with metabolic pathways being the most significant (Figure 5B, Table 3 and Table 4)" (line 243), that TLR9 expression was reduced 4-fold in TLR9-deficient mice is nowhere documented in Figure 5B, Table 3 and 4. Please provide the missing body of evidence.

57) Although the authors claim that "Pathway analysis revealed only two gene pathways with p<0.05 (Table 4) with the axon guidance as the most significant pathway downregulated in PA-treated TLR9 deficient lung macrophages" (line 248), significance is not shown in Table 4. Moreover, the second gene pathway with p<0.05 is not explicitly mentioned. Please fix.

58) Please change "p<0.05" to "P < 0.05" (line 248).

59) Please change "palmitic acid" to "PA" (lines 254, 256, 275, 276, 279, 281, 282, 287).

60) Please replace "acid. ." with "acid." (line 257).

61) It is not exactly clear what the authors mean by "Cre [–]" in "Pathways of genes altered by palmitic acid (vs. BSA control) in wild-type (Cre [–]) mouse lung macrophages" (line 258)?

62) Please change "n=8" to "N = 8" (2x) and "n=5" to "N = 5" (2x) in Tables 2 and 4.

63) Please format "Log2 fold change" (Up-regulated genes) and "Genes" (2x), "Log2 fold change" (2x) (Down-regulated genes) using bold in Tables 2 and 4.

64) Please align "Plxna2", "Plxna4", and "Plxnb1" (Up-regulated genes, Axon guidance) horizontally so that they become exactly centered above all other entries in the "Genes" column of Table 2.

65) Please align "0.5" (Up-regulated genes, Axon guidance) horizontally so that it becomes exactly centered above all other entries in the "Log2 fold change" column of Table 2.

66) Please remove the empty row between "Metabolic pathways" and "Genes", "Log2 fold change", "Genes", "Log2 fold change" (Down-regulated genes) in Table 2.

67) From the legend to Table 2 is not clear why is "C1galt1c1" underlined?

68) Please remove underline formatting from " " in "C1galt1c1 " in Table 2.

69) Please align "Plxna2", "Plxna4", and "Plxnb2" (Down-regulated genes, Axon guidance) horizontally so that they become exactly centered above all other entries in the "Genes" column of Table 4.

70) Please align "-0.6", "-0.2", "-1.1", and "-0.8" (Down-regulated genes, Axon guidance) horizontally so that they become exactly centered below the first entry in the "Log2 fold change" column of Table 2.

71) "Palmitic acid is a major saturated fatty acid in human body, which increased in an obese condition" might not be 100% correct since it is hard to imagine that a "fatty acid" can "increase".

72) Please replace "increased" with "increases" (line 276).

73) Please change "fewer" to "few" (lines 286, 299).

74) Please provide figure reference for "Our data showed that lung neutrophil inflammation with TLR9 deficiency in myeloid cells was about 3-fold lower than that in TLR9 sufficient mice, suggesting a major contribution of TLR9 signaling in myeloid cells to mtDNA-mediated lung inflammation" (line 294).

75) Please change "regulates" to "regulate" (line 310).

76) Although the authors provide rationale for the reduction of the expression level of "Arginase 2" by PA in "Arginase 2 has been shown to promote the resolution of inflammation in lung macrophages" (line 324), there seems to be no word on "Mt-ATP6". Please provide a brief statement on the implications of PA-stimulated Mt-ATP6 downregulation in the Discussion section.

77) Please replace "left in BAL fluid" with "found in BALF" (line 328).

78) Please change "explored in our future experiments" to "explored" (line 337).

79) Please format "National Institutes of Health (NIH): R01AI150082, R01AI152504, R01AI161296, and U19AI125357 to Hong Wei Chu for performing cell culture and mouse model studies." (line 345) consistently with the rest of the text.

80) Please reformat the "Author Contributions" section according to the official IJMS "Instructions for Authors" guidelines.

81) Please replace "manuscript.Niccolette" with "manuscript. Niccolette" (line 349).

82) Please change "manuscript.Hong" to "manuscript. Hong" (line 350).

Author Response

Please see my responses in the attached .pdf file.

Thank you.

Round 2

Reviewer 1 Report

The authors have addressed my concern accordingly.